# Rare Defects: Looking at the Dark Face of the Thrombosis

**DOI:** 10.3390/ijerph18179146

**Published:** 2021-08-30

**Authors:** Giovanna D’Andrea, Maurizio Margaglione

**Affiliations:** Genetica Medica, Dipartimento di Medicina Clinica e Sperimentale, Universita’ degli Studi di Foggia, 71122 Foggia, Italy; maurizio.margaglione@unifg.it

**Keywords:** thrombosis, genetics, anticoagulant, genetic risk, prothrombotic

## Abstract

Venous thromboembolism (VTE) constitutes a serious and potentially fatal disease, often complicated by pulmonary embolism and is associated with inherited or acquired factors risk. A series of risk factors are known to predispose to venous thrombosis, and these include mutations in the genes that encode anticoagulant proteins as antithrombin, protein C and protein S, and variants in genes that encode instead pro-coagulant factors as factor V (FV Leiden) and factor II (FII G20210A). However, the molecular causes responsible for thrombotic events in some individuals with evident inherited thrombosis remain unknown. An improved knowledge of risk factors, as well as a clear understanding of their role in the pathophysiology of VTE, are crucial to achieve a better identification of patients at higher risk. Moreover, the identification of genes with rare variants but a large effect size may pave the way for studies addressing new antithrombotic agents in order to improve the management of VTE patients. Over the past 20 years, qualitative or quantitative genetic risk factors such as inhibitor proteins of the hemostasis and of the fibrinolytic system, including fibrinogen, thrombomodulin, plasminogen activator inhibitor-1, and elevated concentrations of factors II, FV, VIII, IX, XI, have been associated with thrombotic events, often with conflicting results. The aim of this review is to evaluate available data in literature on these genetic variations to give a contribution to our understanding of the complex molecular mechanisms involved in physiologic and pathophysiologic clot formation and their role in clinical practice.

## 1. Introduction

Venous thromboembolism (VTE) is an important disease that represents a major health problem worldwide with an incidence in the general population of 1 per 1000 adults, and a significant different prevalence among distinct ethnic groups, as well as across the ages [1]. General population and hospital-based samples involving individuals of European ancestry (Caucasians) indicated that the incidence of first-time symptomatic VTE varies from 104 to 183 per 100,000 person-years. In comparison to Caucasians, higher rates of VTE are recorded in Africans while lower rates—in Asians.

The susceptibility to VTE is largely accounted for by the clustering of several, possibly inherited, risk factors. Studies of twins and families show that VTE is highly heritable and follows a multifactorial non-Mendelian inheritance model, involving interaction with clinical risk factors. Both acquired and inherited factors play important roles in the pathogenesis of VTE. However, the risk varies greatly from one individual to another, and the causes for many cases remain unidentified. Indeed, also in populations from different ancestries an important proportion of idiopathic VTE events is recorded, the rate of cases ranging from 19% to 40% [2].

Among environmental factors are blood stasis, plasma hypercoagulability, endothelial dysfunction, advancing age, male sex, obesity, surgery, trauma, cancer, immobilization, and pregnancy and use of exogenous hormones [1]. All these risk factors altogether explain almost half of all episodes of VTE. On the other hand, inherited thrombotic factors have been identified in gene variants of transcripts with an antico-agulant function, such as antithrombin, protein C, and protein S, which are considered to be strong genetic risk factors [3]. In addition, variants in genes encoding pro-coagulant factors as FV (in particularly the so-called “FV Leiden”) and FII (prothrombin G20210A) have also been identified, which are, instead, considered moderate genetic risk factors [4].

In a multifactorial model, the susceptibility does not change drastically with the presence or the absence of a specific risk factor, being accounted for by the clustering of several, possibly inherited, risk factors. None of these factors is either necessary or sufficient for the occurrence of VTE, but the totality makes VTE development more likely. This means that the liability to develop VTE is continuously distributed in the general population because of the additive effects of genetic and environmental factors. Our current thinking is that only in individuals whose liability exceeds a certain threshold will VTE then occur.

In the last years, the availability of powerful technologies allowed for a series of studies aimed at investigating novel VTE loci. Several loci have been suggested to affect the risk for VTE, most of them through gene expression in blood and liver tissue [1].

The above-mentioned model was demonstrated to be effective, using 31 replicated common genetic variants [5]. Loci acted in concert each other in an additive fashion to produce the final phenotype. A meta-analysis of genome-wide association studies found suggestive evidence for an association of a large series of com-mon gene variants and VTE [6]. However, uncommon and familial rare variants (e.g., those in antithrombin), even those with large effect sizes, are not likely to be detected using this approach.

Clinical characteristics of patients carrying a strong risk factor are significantly different from people with a moderate risk factor. Usually, in subjects carrying a strong genetic risk factor, VTE occurs at a younger age as an idiopathic or provoked episode, a family history of VTE is often present, and there is a high probability of recurrence [3]. However, in about one-third of patients with a positive family history for VTE or with recurrent episodes, despite our knowledge, the underlying molecular mechanisms remain unsolved [7].

Several additional genetic factors within the procoagulant, anticoagulant, and fibrinolytic pathways were suggested to be potential risk factors for VTE [8].

The objective of this review is to collect the current genetic and clinical knowledge of rare variants in pro-coagulant (fibrinogen, FII, FV, FVII, FXI, tissue factor), anticoagulant (thrombomodulin, ADAMTS13), and fibrinolytic (PAI-1) genes to better clarify VTE pathophysiology and the role of these variants as well as the clinical utility for developing new strategies for the management of VTE patients (Table 1).

###  1.1. Inherited Disorders of Fibrinogen and Thrombosis: Afibrinogenemia and Dysfibrinogenemia

Hereditary fibrinogen abnormalities a prevalence rate of ~8% among the rare bleeding disorders with an estimated prevalence of one in a million [9]. Inherited disorders of fibrinogen encompass two classes of plasma fibrinogen defects: Type I, afibrinogenemia or hypofibrinogenemia, which has a complete absence or low plasma fibrinogen antigen levels (quantitative fibrinogen deficiencies), and Type II, dysfibrinogenemia or hypodysfibrinogenemia, which shows normal or reduced antigen levels but low or very low functional activity (qualitative fibrinogen deficiencies) [10].

#### 1.1.1. Afibrinogenemia

Afibrinogenemia is associated with mild-to-severe bleeding, whereas hypobrinogenemia are most often asymptomatic [10]. In afibrinogenemic patients venous and arterial thrombotic events have been rarely reported [11]. Thrombotic events can occur spontaneously in half of the cases, or with concomitant risk factors such as surgery procedures [11], replacement therapies such as fibrinogen concentrate [12] or recombinant factor VIIa plasma infusion [13]. Furthermore, in young afibrinogenemic women, with a positive family history of thrombosis an increased rate of the occurrence of thrombotic events, unrelated to replacement therapy, has been found [14]. However, in the majority of patients, no known risk factor was present. Many hypotheses have been proposed to explain the predisposition to thrombosis of afibrinogenemic subjects. Based on animal models it has been shown that in the absence or reduction of fibrinogen, the αIIbβ3 receptor, which normally transfers the fibrinogen molecule into the platelet α-granule, imports fibronectin. In turn, the αIIbβ3-bound fibronectin may promote platelet aggregation [15]. Furthermore, in the absence of fibrinogen platelet aggregation is possible due to the action of von Willebrand factor (VWF), which stabilizes the factor VIII and interacts with subendothelial components and platelet membrane receptors [15]. On the basis of the mechanism described above, VWF and fibronectin might play a compensatory role in the absence of fibrinogen. Indeed, fibrinogen absence increases the levels of other procoagulant factors, and this could increase patients’ prothrombotic susceptibility, with a tendency to embolize [15]. In any case, available data in the literature suggest that afibrinogenemia is associated with thromboembolic complications with or without fibrinogen replacement therapy [14]. In conclusion, although thrombosis is very rare in afibrinogenemic patients, a series of episodes were reported in the literature. However, considering the conflicting studies carried out to date, the pathophysiology of thrombotic events in these patients is largely unknown and needs to be further investigated.

#### 1.1.2. Dysfibrinogenemia

Dysfibrinogenemia is rare, with a prevalence of approximately 15 per 100,000 people, and clinical manifestations have a high phenotypic variability [16]. Indeed, most of them are clinically asymptomatic, whereas some can present with bleeding diathesis or thrombotic episodes, rarely with both [16]. Bleeding episodes involve mostly the skin and mucus membranes, less frequently the musculoskeletal apparatus, genitourinary and gastrointestinal tract [17,18]. On the other hand, most severe and often fatal bleeding episodes may involve the central nervous system with intracranial hemorrhage [19]. Women with dysfibrinogenemia, in some cases have obstetric complications, such as recurrent abortions and pre- and post-partum bleeding episodes [20]. Although many patients with dysfibrinogenemia bleed, both arterial and venous thromboembolic disease have been observed [21,22]. Thrombotic events can occur in the presence of concomitant risk factors such as co-inherited thrombophilic risk including natural inhibitors abnormalities, such as factor V Leiden, or acquired risk factors, pregnancy, and replacement therapy [11]. However, in many patients, no known genetic or acquired risk factors were found. Recently, patients with chronic thromboembolic pulmonary hypertension (CTEPH) in association with dysfibrinogenemia have been described. In these patients, an increased resistance to the fibrinolytic system, as a consequence of a change in the molecular structure of the fibrin, leading to the development of CTEPH after acute thromboembolism has been suggested [23]. Two hypotheses that can explain why individuals can develop thrombotic events in cases of dysfibrinogenemia have been proposed. First, the abnormal fibrinogen may have a defective binding to thrombin, with a consequent increase of the latter. Secondly, the fibrin clot containing a dysfunctional fibrinogen may be less sensitive to plasmin during tissue-type plasminogen activator (t-PA)-mediated fibrinolysis [24]. In addition, an abnormal fibrinogen molecule can display abnormal interactions with platelets, impaired assembly of the fibrinolytic system, and abnormal calcium binding that affect the polymerization pocket [25].

#### 1.1.3. Prothrombin

Variants in the prothrombin gene (F2) causing abnormally low factor II (FII) antigenic and functional levels (Type I defects) have always been associated with bleeding, often severe bleeding [26]. In addition, different gene variants, causing low FII activity but normal or near normal antigen levels, show a milder bleeding tendency. Recently, a series of studies identified unrelated families carrying different variants in the same FII codon (p.Arg596) encoding for different amino acids (Leu [27]; Gln [28]; and Trp [29]), which causes a resistance to the anticoagulant effect of antithrombin and a consequent thrombophilic state. Findings of rare additional patients carrying the p.Arg596Leu variant further confirm the possibility that F2 may harbor rare variants strongly associated with VTE.

#### 1.1.4. Factor V (FV)

In addition to the frequent factor V (FV) gene (F5) variant p.Arg506Gln, also known as FV Leiden, the most frequent inherited VTE risk factor among Caucasians, few rare, or quite rare F5 variants have been described to be associated with an increased risk for VTE [30,31,32]. Variants occurring at a cleavage site for anticoagulant activated protein C (p.Arg306Thr and p.Arg306Gly), a key event for the inactivation of activated FV, resulted in partial FVa inactivation [33]. However, the effect on VTE is limited, if any. Private F5 gene variants associated with VTE were described. The FV p.I359T substitution was found to confer a risk for VTE through resistance to activated protein C, but only when co-inherited with a F5 null allele.

More recently, two different homozygous substitutions (p.Trp1920Arg and p.Ala2086Asp) [34,35], both affecting FVa inactivation, were identified and further showed the FV as a target for innovative anticoagulant drugs.

#### 1.1.5. Factor VII (FVII) Deficiency

The FVII deficiency shows a high genetic and clinical heterogeneity. Although gene variants within the F7 gene are essentially responsible for a bleeding phenotype, in the last years, FVII deficiency was suggested to causes both venous and arterial thrombosis [36]. Thrombotic episodes, particularly VTE, have been re-ported in 3% to 4% patients carrying a FVII deficiency, even in those presenting with a severe deficiency [37]. In patients with FVII deficiency, arterial thromboses are less common than VTE, the latter being described in unusual locations such as central retinal and cerebral veins, as well as portal and splenic veins [38]. Thrombotic episodes involve different system and can occur spontaneously or in conjunction with treatment or others predisposing factors [36]. However, spontaneous thrombotic events are very rare and mainly occur after childbirth, main surgery, replacement therapy (particularly with rFVIIa), or in association with other thrombophilic risk factors such as FV Leiden, prothrombin G20210A variant, elevated FVIII levels, and antiphospholipid antibodies [36]. Pathogenic mechanisms leading to thrombotic events are not yet fully understood. A specific role for particular F7 variants was suggested. Indeed, two gene variants, p.Arg304Gln (FVII Padua) and p.Ala294Val were highly represented in patients with FVII deficiency and thrombosis events [39]. Both mutations give rise to a type 2 defect, patients presenting low activity but normal or slightly reduced FVII antigen and are associated with mild clinical thrombotic phenotype [40]. Indeed, thrombotic events have been described only in two cases of “true” or type I deficient patients [41]. The pathogenetic mechanism induced by p.Arg304Gln and p.Ala294Val gene variants is the consequence of an amino acid substitution in the catalytic domain of the molecule, which probably affects the interaction of FVII with TF. Biochemical studies showed that both variants induce a conformational change of a β-Strand in the FVIIa catalytic domain that empowers its binding capacity with the TF [38]. The following modification of the ability to prime the extrinsic coagulation pathway in vivo, further stresses the critical role of the TF:VIIa complex in the blood coagulation cascade [38].

#### 1.1.6. Factor FXI (FXI)

Factor XI (FXI) is the zymogen of a plasma protease, factor XIa (FXIa), that contributes to thrombin generation during blood coagulation by proteolytic activation of several coagulation factors, most notably FIX [42]. FXI deficiency was first described in literature in 1953, it is a rare genetic bleeding disorder caused by reduced levels and activity of FXI clotting factor [42]. The clinical phenotype of individuals with FXI deficiency is highly variable and not at all correlated with FXI antigen levels and activity, therefore complicating the ability to predict the bleeding phenotype [42,43]. In the past, on the basis of animal models and clinical studies, FXI deficiency was essentially associated with a protective effect from thrombotic events [44], due to a reduction of thrombin generation and a weaker stability of blood clot [44]. The occurrence of VTE seems mainly be associated with the replacement therapy, particularly after infusion of concentrated plasma-derived FXI [45]. Indeed, a significant lower VTE incidence in patients with severe FXI deficiency was found, further stressing the protective role of low FXI levels [46]. On the other hand, in several studies higher FXI levels were associated with an increased risk for ischemic stroke [47] and arterial thrombosis [48], while the role for FXI in myocardial infarction (MI) is less clear [49]. The association of high FXI levels with the risk of a first VTE event was investigated [50]. Subjects with FXI levels in the upper quartile of the distribution (above 110 U/dL) had a twofold increased risk as compared to subjects with FXI levels in the lowest quartile (below 83.3 U/dL). However, these data were not confirmed [51]. In FV Leiden carriers, high FXI levels were suggested to contribute to the risk for VTE [52]. Furthermore, animal models al-so suggest that the activation of FXI by FXIIa promotes pathological thrombus formation [53]. FXI inhibition has been proposed as an innovative therapeutic tool to reduce the risk for VTE [54]. Recent studies confirmed the potential use of FXI inhibition for the prevention of VTE [55,56]. Thus, current evidence is strong enough to support the role of high FXI levels as a risk factor for VTE.

## 2. Tissue Factor Pathway Inhibitor (TFPI)

Tissue factor (TF) is best known as the primary cellular initiator of blood coagulation. After vessel injury, the TF:FVIIa complex activates the coagulation protease cascade, which leads to fibrin deposition and activation of platelets, highlighting its fundamental role in the hemostatic process [57]. TF expression by non-vascular cells plays an essential role in hemostasis by activating blood coagulation. In contrast, TF expression by vascular cells induces intravascular thrombosis [58]. In the last two decades, moreover, TF was described as a glycoprotein located in several tissue including vascular wall and atherosclerotic plaque and circulates in the blood associated with microparticles (Mps) [59]. TF is a “true surface receptor” involved in many intracellular signaling, cell-survival, gene and protein expression, proliferation, angiogenesis, and tumor metastasis [60]. There is now strong experimental evidence that tissue factor pathway inhibitor (TFPI) is a critical inhibitor that modulates tissue factor-induced coagulation. However, the role of TFPI as a risk factor for thrombosis is yet to be determined. Coagulation inhibitors play important roles in preventing individuals from thrombosis and, a limited level of evidence suggests that low plasma TFPI levels are associated with ischemic stroke or venous and arterial thrombotic disorders [61]. Furthermore, a predictive risk value for VTE and tumor metastasis was also associated to TF, this marker displaying a high diagnostic sensitivity and specificity [62]. Plasma TFPI levels were significantly decreased in patients with thrombotic thrombocytopenic purpura (TTP) compared with those in healthy volunteers [63]. Recently, it has been shown that total TFPI was higher in individuals with higher procoagulant factor levels and advancing age [64]. Only subjects with TFPI levels below the 5th percentile (<18.8 ng/mL) showed a modest increase of the risk for VTE, after adjusting for procoagulant factor levels [64]. Despite a series of animal models, very few clinical studies were performed in humans to date. Then, current knowledge is not sufficient to show whether TFPI levels affect the risk for VTE. Based on the available results, there is no clinical evidence to measure TF. Further ad hoc clinical trials are needed to assess whether higher TF or low TFPI levels are a risk factor for VTE.

### 2.1. Thrombomodulin

Thrombomodulin (TM) is the endothelial cell cofactor for protein C activation working as a modulator of coagulation system and inflammation. TM enhances thrombin-catalyzed activation of protein C. Activated protein C (APC) proteolytically inactivates blood coagulation factors Va and VIIIa [65]. Normal APC generation depends on the precise coupling of thrombin and protein C to their respective receptors, TM and endothelial protein C receptor (EPCR) on the surface of endothelial cells [66]. The thrombin-TM complex is also involved in the physiological regulation of fibrinolysis by activating thrombin-activated fibrinolysis inhibitor (TAFI) [67]. Considering both anticoagulant and pro-fibrinolytic activities of the thrombin-TM-PC-EPCR system, antithrombotic functions are evident. Animal model data suggest that TM dysfunction or deficiency may be associated with a prothrombotic disorder [68,69]. Recently, transgenic mice with a missense mutation in the TM gene (THBD) corresponding to human E387P, developed a prothrombotic disorder showing an improvement in fibrin deposition [70], probably due to the thrombomodulin inability to catalyze in vitro thrombin activation of protein C to APC. The mutant mouse strain was able to repeat different mechanisms of thrombotic events in humans, confirming previous knowledge and suggesting new hypotheses to be developed in clinical studies.

In patients with VTE, a series of clinical studies addressed the role of sporadic mutations and polymorphisms in the TM gene [71]. In these studies, genetic variation in the promoter, coding region, and 3′-untranslated regions (3′-UTR) of the TM gene were considered [72,73]. Although some studies suggested a role for THBD variants, results are conflicting. In addition, genome-wide studies also gave inconsistent findings [73]. The association between a disease and a gene variant arises because the latter is directly causative or is in strong association with a causative variant. The occurrence of several factors may explain inconsistencies among studies that addressed the identification of THBD disease-causing variants, including insufficient statistical power, population stratification, various forms of between-study heterogeneity, including differences in genetic ancestry, ascertainment schema, environmental influences, and time-varying associations [73,74]. Although it is difficult to known to what extent mutations in the thrombomodulin gene can actually modify its function as modulator of the coagulation and fibrinolysis pathways, it is conceivable that any impairment of TM exposure on the plasma membrane can produce a lower thrombin binding. The following increased amounts of unbounded thrombin are expected to induce an imbalance between its procoagulant (increased fibrin formation) and anticoagulant (impaired APC generation) properties. Indeed, the bleeding phenotype in patients carrying a THBD mutation associated with increased soluble protein levels and decreased thrombin generation further stresses the pivotal role of THBD as key regulator [75].

### 2.2. Disorders of Fibrinolysis

The fibrinolytic system involves the conversion of plasminogen (PLG) into plasmin by the action of tissue plasminogen activator (tPA) and urokinase plasminogen activator (uPA). The plasminogen activator inhibi-tor-1(PAI-1) is a serine protease inhibitor that plays an important role in the regulation of fibrinolysis. PAI-1 play a role in the inhibition of the activity of tPA and uPA [76]. Additional inhibitors of the fibrinolytic pathway are TAFI and α2-antiplasmin [77]. PLG deficiency is a rare disorder that has been classified as type I or hypoplasminogenemia, and type II or dysplasminogenemia [78]. In type I, both plasminogen activity and antigen levels are decreased. In contrast, type II is characterized by decreased plasminogen activity but normal antigen levels. A series of mutations has been described in the PLG gene that cause plasminogen deficiency, such as missense, nonsense, frameshift, splice site, deletion, and insertion mutations [79]. These studies suggested that the most common molecular defects are in association with type I [80]. How-ever, there is not an association between the genotype, the number or the type of putative pathogenic mutations in PLG with the occurrence of VTE [81]. These data suggest that decreased plasminogen levels do not, in and of themselves, increase the risk of VTE. Plasminogen activator inhibitor-1 (PAI-1) deficiency, causing enhanced fibrinolysis due to the decreased inhibition of plasminogen activators, results in an in-creased conversion of plasminogen to plasmin, excessive levels of plasminogen activator inhibitor-I (PAI-I) and hyperfibrinolysis [78]. Plasma hypofibrinolysis was shown to be associated with an increased risk for VTE [82]. This risk was explained by elevated plasma levels of TAFI and PAI-1. These events are causally related to the development of atherosclerosis and associated to thrombotic complications, as well as with the development of venous and arterial thrombosis. Indeed, plasma concentration of plasminogen tends to increase 5–10 times during arterial or vascular disorders and this would explain its role in the pathogenesis of venous and arterial thrombosis [83]. Many studies have associated to PAI-1 the function of mediator of organ fibrosis emphasizing further its involvement in the pathogenesis of atherosclerosis [83]. A PAI-1 gene polymorphism in the promoter region (4G/5G) was the one most widely studied. Indeed, the 4G allele has been found to be associated with rising of plasma PAI-1 levels in different ethnic populations. However, the association with VTE is still controversial [84]. In a cohort of unselected patients, it has been found a significant association between the 4G/5G polymorphism and the risk of VTE or coronary artery disease [85]. However, in most of patients, there was the contemporary presence of Factor V Leiden or others pre-disposing risk [86]. Furthermore, other studies have shown that the 4G allele is associated only with a modest increase of VTE risk, particularly in subjects with other genetic thrombophilic defects [87], thus only representing a weak additional risk. Then, the role of fibrinolytic pathway in pathogenesis of VTE re-mains, to date, unclear and, consequently, more investigations are needed.

## 3. ADAMTS13

ADAMTS13 is a metalloproteinase synthesized predominantly in the liver and responsible for the modulation of the molecular size of von Willebrand factor (VWF) multimers in plasma. The ADAMTS13 role consists of cleaving VWF ultra-large molecules, initiating platelet binding to sub-endothelial surface and subsequent platelet adhesion [88]. The association between ADAMTS13 dysfunction and thrombotic thrombocytopenic purpura (TTP) [88,89], diabetes [90], pre-eclampsia [91], and acute myocardial infarction [92] has been well documented. TTP is a severe microangiopathic disorder of the blood-coagulation system. A series of animal models demonstrated the role of ADAMTS13 in the pathogenesis of ischemic stroke [93]. ADAMTS13 down-regulates both thrombosis and inflammation through cleavage of VWF multimers. Dysfunction of the ADAMTS13-VWF leads to large VWF multimers accumulation, leukocyte rolling and adhesion of platelets, which is the first step of thrombosis and inflammation [94]. Furthermore, a VWF multimers accumulation increases plasma factor VIII (FVIII) levels, a known independent risk factor for VTE [95]. Deficiency in ADAMTS13 endopeptidase contributes to the development of VEGF inhibitor-related thrombotic microangiopathies. In addition, patients with sepsis or DIC have decreased levels of ADAMTS13 and increased levels of VWF [96], which play a key role in initiating thrombus formation. Complete deficiency in ADAMTS13 induces a prothrombotic state, which represents an important risk factor for TTP or stroke, but it is insufficient to cause TTP or stroke by itself. It may result in ischemic stroke in conjunction with additional genetic or environmental factors [97]. Considering the pivotal role of ADAMTS13 in the modulation of the coagulation cascade, a significant effect in the development of VTE has been suggested [98]. It was supposed that genetic factors affecting ADAMTS13 activity might modulate the risk for VTE by modifying VWF and FVIII levels. In patients with VTE, the association with sporadic and common gene variant has been investigated [99]. However, no polymorphism was found to correlate with VTE. Recently, using a next generation sequencing (NGS) approach in patients with idiopathic VTE, rare and low-frequency gene variants of the ADAMTS13 coding region were found with a significantly higher frequency [100]. These findings suggested that genetic variants modulating the VWF-cleaving activity may modulate the risk for thrombosis. To date, studies investigating ADAMTS13 activity as a result of molecular abnormality and its ability to predict thrombosis events produced contradictory results. Then, further studies in patients with low level of ADAMTS 13 and rare or common gene variants are needed in order to perform a more accurate genotype-phenotype correlation to understand the role on arterial and venous thrombosis pathogenesis.

## 4. Conclusions

Even though many years of investigation aimed at improving diagnosis and management, VTE remains a significant public health threat. In general population, VTE heritability has been estimated to account for approximately 30% [101], whereas a higher figure was shown between twins (50%) [102].

The heritability of VTE is very high and patients often report positive family history of VTE. Efforts to predict and prevent venous thromboembolic disease largely depends on our ability to accurately identify patients at risk. On the whole, known gene variants only explain a small portion of the VTE heritability. Thus, many of the disease-causing mechanisms underlying VTE remain to be fully characterized, including those involving genetic risk factors. Although genes encoding for protein involved in the coagulation pathway are natural candidate genes for VTE, genome-wide association studies found suggestive evidence for a role of genes involved in different pathways, e.g., encoding for platelet and red blood cell traits. This review provided an overview of available data to date on studies that investigated inherited venous thrombosis, rare gene variants, as potential thromboembolic risk factors. To date, the clinical utility in diagnostic practice of specific tests for the investigation of these rare defects in VTE patients is unknown. A better understanding of the wide-ranging and complex role of these disorders in both thrombosis and hemostasis will allow for a better prediction of the thrombotic risk in the general population as well as in different clinical settings of patients. The identification of new candidate genes is urgently needed to improve current risk prediction models for VTE. Moreover, the identification of genes with rare variants but a large effect size may pave the way for studies addressing new antithrombotic agents, in order to improve the management of VTE patients.

Cutting edge technologies may enable a thorough and clever estimation of the personal risk profile, which is currently relatively inaccurate, involving the assessment of “omics” signatures or biomarkers. This approach will represent a welcome improvement in the ability to measure the risk for VTE in a meaningful way.

## Figures and Tables

**Table 1 ijerph-18-09146-t001:** Characteristics of thrombosis of congenital disorders examined.

Congenital Defects	Type of Thrombosis	References
Afibrinogenemia	Arterial and mainly venous	Korte W. et al., 2016
Dysfinogenemia	arterial and mainly venous	Korte, W. et al.; 2016
Phrotrombin	mainly venous	Djordjevic, V, 2013; Bulato, C. 2016
FV	mainly venous	Chan, WP 1998; Mumford, AD 2013
FVII	mild arterial risk, mainly venous	Marty, S. 2008.
FXI deficiency	mild venous, mainly in association with replacement therapy	Palla, R. 2015; Puy, C. 2016.
High FXI level	arterial (conflicting results)	Suri MF, 2010.Yang DT, 2006. Meijers JCM, 2000. I M Rietveld, 2019.
TFPI	Athersclerotic plaque, intravascular thrombosis conflicting results, predictive value in tumor metastasis	Toschi, V. 1997; Engelman, B. 2003.
Trombomodulin	Animals model, conflicting results in humans	Ahmad, A. et al. 2017; Burke, J.P. et al. 2005.
PAI-1 deficiency	Mainly venous	Meltzer, M.E. et al. 2010.
High plasminogen level	Arterial and venous	Flevaris, P. 2017; Margaglione, 1998.
ADAMTS13	TTP conflicting results for arterial and venous	Levy, G.G. 2001; Akyol, O. 2015; Xin, C. 2019; Bittar, L.F. 2010; L.A. Lotta, 2013.

## Data Availability

Not applicable.

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
