# Peer review of "Rare Defects: Looking at the Dark Face of the Thrombosis"

_ijerph, 2021, doi:10.3390/ijerph18179146_

Round 1

Reviewer 1 Report

The manuscript entitled: “Rare defects: looking at the dark face of the thrombosis” by Giovanna D’Andrea and Maurizio Margaglione told about the importance of genetic variants in the molecular mechanisms involved in Venous Thromboembolism (VTE).

Considerations

The novelties of the study are not clear, despite having a well-defined objective, it was partially contemplated, since the information present in the conclusion is quite comprehensive and exploratory, focused only point of the hereditary factors that lead to VTE development.

The nomenclature is not clear, what is the most usual, pro coagulant or prothrombotic? Do they have the same meaning? Standardization and better definition about VTE is recommended.

Vascular, cardiovascular and related diseases are, in fact, the main cause of morbidity and mortality worldwide, but the references used to support this statement may be out of date (2007), it is recommended to use more recent epidemiological data.

The data presented and discussed are quite descriptive and also speculative, lack of experimental data and current significant statisticians (within the possibilities) that better support the raised questions.

Define and discuss more clearly the main differences between afibrinogenemia and dysphbrinogenemia in VTE.

Despite bringing many important data present in the literature, the theoretical reference of this study, is sometimes conflicting and inconclusive, which makes it difficult to discuss and understand the subject, therefore, what contributions regarding the heritability, genetic variants or VTE in general, has your research group been bringing?

What other investigation strategies would you use in order to elucidate the mechanisms involved in identification of new candidate genes to improve current risk prediction models for VTE?

Issues that could be addressed in the discussion of this review: What approaches regarding advances in the investigation of signaling pathways and cellular mechanisms, involving or not the use of promising drugs in the treatment and/or prevention of VTE, could be investigated?

What are the scientific and technological innovations already available for the treatment and improvement of prognosis aiming at increasing the quality and life expectancy of patients with VTE?

Reviewer 2 Report

Review Report: Title Rare defects: looking at the dark face of the thrombosis Authors Giovanna D'Andrea * , Maurizio Margaglione

in the paragraph,
"Dysfibrinogenemia
The frequency of dysfibrinogenemia is approximately 8 per 1,000 individuals."

I suggest looking in other sources for confirmation of these incidence values.
